# Machine learning with random subspace ensembles identifies antimicrobial resistance determinants from pan-genomes of three pathogens

**Jason C. Hyun**[1], **Erol S. Kavvas**[2], **Jonathan M. Monk**[2]*, **Bernhard O. Palsson**[2]*

**1** Bioinformatics and Systems Biology Program, University of California, San Diego, La Jolla, California, United States of America, **2** Department of Bioengineering, University of California, San Diego, La Jolla, California, United States of America

\* jmonk@ucsd.edu (JMM); palsson@ucsd.edu (BOP)

**Data Availability Statement:** All genome sequences and their associated metadata used in this study are available on the PATRIC database (https://www.patricbrc.org/). Genome IDs for

## Abstract

The evolution of antimicrobial resistance (AMR) poses a persistent threat to global public health. Sequencing efforts have already yielded genome sequences for thousands of resistant microbial isolates and require robust computational tools to systematically elucidate the genetic basis for AMR. Here, we present a generalizable machine learning workflow for identifying genetic features driving AMR based on constructing reference strain-agnostic pan-genomes and training random subspace ensembles (RSEs). This workflow was applied to the resistance profiles of 14 antimicrobials across three urgent threat pathogens encompassing 288 *Staphylococcus aureus*, 456 *Pseudomonas aeruginosa*, and 1588 *Escherichia coli* genomes. We find that feature selection by RSE detects known AMR associations more reliably than common statistical tests and previous ensemble approaches, identifying a total of 45 known AMR-conferring genes and alleles across the three organisms, as well as 25 candidate associations backed by domain-level annotations. Furthermore, we find that results from the RSE approach are consistent with existing understanding of fluoroquinolone (FQ) resistance due to mutations in the main drug targets, *gyrA* and *parC*, in all three organisms, and suggest the mutational landscape of those genes with respect to FQ resistance is simple. As larger datasets become available, we expect this approach to more reliably predict AMR determinants for a wider range of microbial pathogens.

## Author summary

Antimicrobial resistance remains a persistent threat to global public health, with 700,000 deaths each year attributable to resistant bacterial infections. The falling cost of genome sequencing offers an avenue for rapidly predicting and elucidating the resistance profiles of infectious isolates, which is necessary for the design of more effective antimicrobial therapies from existing drugs. As such, clinical surveillance programs have already yielded sequences for thousands of distinct, resistant strains of most major pathogens. Here, we have developed a workflow for training machine learning models capable of not just predicting resistance profiles from genome sequences, but also

specific strains used are available in S1 Dataset. All reference sequences used for identifying antimicrobial resistance genes are available in S2 Dataset.

**Funding:** This research was supported by a grant from the National Institute of Allergy and Infectious Diseases (AI124316, awarded to JMM and BOP, https://www.niaid.nih.gov/). This research was also supported by a grant from the National Institutes of Health (T32GM8806, awarded to JCH, https://www.nih.gov/). The funders had no role in study design, data collection and analysis, decision to publish, or preparation of the manuscript.

**Competing interests:** The authors have declared that no competing interests exist.

identifying the responsible genes. When applied to 14 drugs and three urgent threat pathogens (*Staphylococcus aureus*, *Pseudomonas aeruginosa*, and *Escherichia coli*), our approach outperformed common statistical methods for detecting gene-level associations, identifying a total of 45 known resistance-conferring genes, as well as 25 candidate genes potentially involved in new mechanisms of resistance. These results show that this method can generalize to other drugs and pathogens to predict and explain resistance profiles at the gene level.

## Introduction

The emergence of antimicrobial resistance (AMR) remains a persistent problem in the treatment of bacterial infections. Since the discovery of penicillin in 1928, pathogens have developed resistance to almost all major antibiotics, often within a few years of their introduction [1, 2]. Advancements in sequencing technology have already yielded hundreds to thousands of publicly-available genome sequences for each major bacterial pathogen [3], and analyzing this deluge of data will require robust analytic workflows to extract insights on the acquisition of resistance, its genetic basis, and the underlying molecular mechanisms.

AMR prediction models have already been developed from genome sequence collections of many pathogens, such as *Staphylococcus aureus* [3, 4, 5], *Mycobacterium tuberculosis* [4, 6, 7], *Salmonella* [8, 9], *Klebsiella pneumoniae* [10, 11], and *Neisseria gonorrhoeae* [12, 13]. However, these approaches are often designed to maximize accuracy in predicting AMR phenotypes, emphasizing their diagnostic capabilities over their capacity to uncover genetic mechanisms for resistance. Many such models are also based on the detection of genes from a curated set of known AMR determinants, rendering them difficult to generalize to different treatments or organisms and unsuitable for discovering novel genes or interactions that drive resistance. Continued reductions in sequencing costs will enable whole genome sequencing (WGS) of these pathogens at an increasing scale, and soon expand the capabilities of statistical approaches beyond the prediction of AMR phenotypes and towards the reliable identification of their genetic determinants. Thus, computational tools developed with both goals of predicting and explaining AMR phenotypes are sorely needed.

The identification of gene-AMR relationships falls under the umbrella of microbial genome-wide association studies (GWAS), which bear many similarities to human GWAS [14]. However, microbial GWAS methods are still under development as traditional human GWAS methods struggle to generalize to highly clonal datasets without complex adjustments for population structure [15, 16, 17]. We present here a simple, reference-agnostic, machine learning approach based on pan-genomes for identifying AMR-associated genes using random subspace ensembles (RSEs), previously shown to improve the accuracy of support vector machines trained on high-dimensional biological imaging data [18]. In contrast to more commonly used bootstrapping ensembles, RSEs aggregate classifiers trained on random subsamples of both the sample set (genomes with associated AMR phenotypes) and the feature set (genes and alleles identified in those genomes). We find this method to both accurately predict AMR phenotypes as well as detect known AMR determinants more reliably than well-known association tests or other ensemble strategies, and use this method to predict novel AMR-linked genes for multiple antimicrobials in *S. aureus*, *P. aeruginosa*, and *E. coli*.

## Results

### Selection of genetic features through pan-genome construction

Sets of 288, 456, and 1588 publicly-available genomes for *S. aureus*, *P. aeruginosa*, and *E. coli*, respectively, were downloaded from PATRIC after filtering by contig count and availability of experimental AMR phenotype data (S1 Dataset) [19]. To convert these genome assemblies into fixed feature sets amenable to machine learning, we first constructed a pan-genome for each organism by clustering open reading frames by protein coding sequence into putative genes and classifying each gene as either core (missing in 0–10 genomes), accessory (missing in >10 genomes, present in >10 genomes), or unique (present in 1–10 genomes). This 10-genome threshold was selected by identifying when the core genome size stabilizes as the threshold for core gene was gradually relaxed (S1 Fig). We find that this reference genome-agnostic strategy for gene identification produces pan-genomes consistent with previous pan-genome studies in terms of core-genome size, pan-genome openness, and relationship between gene function and gene frequency (see S1 Text).

Furthermore, as the causative variation responsible for AMR often exists at the level of individual mutations, we identified and enumerated all observed unique amino acid sequence variants or "alleles" of each gene for each pan-genome (S1 Table). Individual genomes were encoded based on the presence or absence of core gene alleles and the presence or absence of non-core genes, yielding a binary matrix representation of genetic variation for each pan-genome that is not biased towards a reference genome and encodes both fine-grained allelic variations in the core genome and broader variations in the dispensable genome.

### Support vector machine ensembles identify known AMR genes more reliably than common statistical tests from the *S. aureus* pan-genome

We focus initially on the *S. aureus* pan-genome to test variations of a recently reported support vector machine (SVM) approach [6], and evaluate their capacity to detect genes from an *a priori* assembled list of known AMR determinants, compared to traditional statistical association tests. We examined six antibiotic treatments against *S. aureus* from distinct drug classes for which experimentally measured AMR phenotype data was available, binarized as Susceptible versus Resistant (S2 Table): ciprofloxacin (fluoroquinolone), clindamycin (lincosamide), erythromycin (macrolide), gentamicin (aminoglycoside), tetracycline (tetracycline), and trimethoprim/sulfamethoxazole (dihydrofolate reductase inhibitor/sulfonamide). For validation, known AMR genes were compiled from literature and the CARD database [20] (S2 Dataset), then aligned to the alleles in the pan-genome using blastp to identify those that were present in our dataset. From an initial query of 915 sequences, we detected 32 unique genes associated with AMR for at least one of the six antibiotics, spanning 304 distinct alleles in the *S. aureus* pan-genome (Table 1). For each allele, the log odds ratio (LOR) for resistance against the corresponding drug and its frequency of occurrence was plotted (S2 Fig). Aside from rare alleles, we find that alleles of genes involved in either active protection of the drug target or inactivation of the drug molecule almost always have large, positive LORs. However, alleles of genes that may confer AMR via a target site mutation or efflux span a wider range of LORs; this may be due to some site mutants not having mutations that directly confer AMR (in which case, large, negative LORs were observed), and some efflux pumps being individually insufficient for conferring clinically relevant levels of resistance.

To define a baseline level of performance for identifying AMR genes from phenotype associations, we examined how reliably common association tests can detect known AMR genes when sorting by p-value. Examining each antibiotic individually, Fisher's Exact and Cochran-

**Table 1. Known AMR genes present in the *S. aureus* pan-genome.**

| Antibiotic | Genes |
|---|---|
| ciprofloxacin | *gyrA* [21,22], *gyrB* [21,22], *parC* [21,22], *parE* [21,22], *norA* [23], *norB* [23], *norC* [23], *sdrM* [23], *mdeA* [23], *qacA* [23], *mepA* [23], *mepR* [23], *mgrA* [23], *arlR* [23], *arlS* [23] |
| clindamycin | *ermA* [24,25], *ermC* [24,25], *lmrS* [26], *linA* [24] |
| erythromycin | *ermA* [24,25], *ermC* [24,25], *lmrS* [26], *msrA* [27], *mphC* [24] |
| gentamicin | *aph(3')-III* [28,29], *ant(4')-I* [28], *aac(6')-aph(2″)* [28,29], *ant(6')-Ia* [30] |
| tetracycline | *tetK* [31], *tetM* [31], *tet38* [32], *norB* [23], *mgrA* [23] |
| trimethoprim | *folA* [33], *dfrA* [33], *dfrG* [34] |
| sulfamethoxazole | *folP* [33] |

Mantel-Haenszel (CMH) tests were applied between each *S. aureus* genetic feature and the AMR phenotypes for that antibiotic, and features were ranked by p-value with fractional ranking to address ties. For the CMH tests, genomes were stratified into clusters generated by applying hierarchical clustering to the genetic feature matrix; the resulting clusters align closely to known subtypes and share similar AMR profiles (Fig 1).

Using the same feature matrix and AMR phenotypes, two types of SVM ensemble were trained for each antibiotic case to classify genomes as susceptible or resistant, composed of 500 SVMs each trained on either 1) a random sample of 80% of genomes and all features to yield a bootstrap ensemble similar to in [6], or 2) a random sample of 80% of genomes and 50% of features to yield a random subspace ensemble (RSE), an adjustment previously shown to improve the accuracy of SVMs trained on high-dimensional biological data (Fig 2a) [18]. Analogously, features were ranked by feature weight (Fig 2b).

We find that both SVM methods consistently identified more known AMR features within both the top 10 and top 50 hits than either statistical test (Fig 2b). For instance, *ermC* and *lmrS* for clindamycin and erythromycin were only detectable by SVM methods, and *aac(6')-aph(2″)* for gentamicin was detected as ranks 1 and 3 by the two SVM methods, compared to much higher ranks 84.5 and 148 by Fisher's Exact and CMH tests, respectively. Additionally, the RSE approach allowed for known AMR genes to be detected at lower ranks compared to bootstrapping in several cases; notably, *lmrS* for clindamycin and erythromycin was detected more than 70 ranks lower with this adjustment, putting *lmrS* within the top 50 hits in both cases with the random subspace approach. To control for phylogenetic distribution, SVM-RSE was also run with either oversampling (SVM-RSE-O) or undersampling (SVM-RSE-U) of genomes to balance the representation of the clusters used in CMH. However, the impact these controls have on the detection of individual known AMR genes is highly variable and does not suggest an improvement overall (Fig 2b). For instance, SVM-RSE-O is the only approach able to identify *ermA* for clindamycin in the top 10, but loses a *gyrA* allele and two *parC* alleles for ciprofloxacin detected by SVM-RSE. Similarly, SVM-RSE-U improves the ranking of several known AMR genes already in the top 10 when compared to SVM-RSE, but loses *lmrS* from the top 50 for both clindamycin and erythromycin and loses *dfrG* for sulfamethoxazole/trimethoprim entirely. Finally, we note that Fisher's Exact test was able to capture two tetracycline resistance genes (*tetM*, *tet38*) albeit at a high ranking of 83.5, while the other three approaches all identified only *tetK* as rank 1 and neither of the other two. However, Fisher's Exact test suffered from an extremely high number of significant hits with Bonferroni correction to FWER $\leq$ 0.05 (S3 Table), most likely due to strong lineage effects driving resistance in which detected features are often markers for a highly resistant subtype rather than true AMR genes [15]. The CMH test with inferred clusters resulted in a more reasonable amount of significant

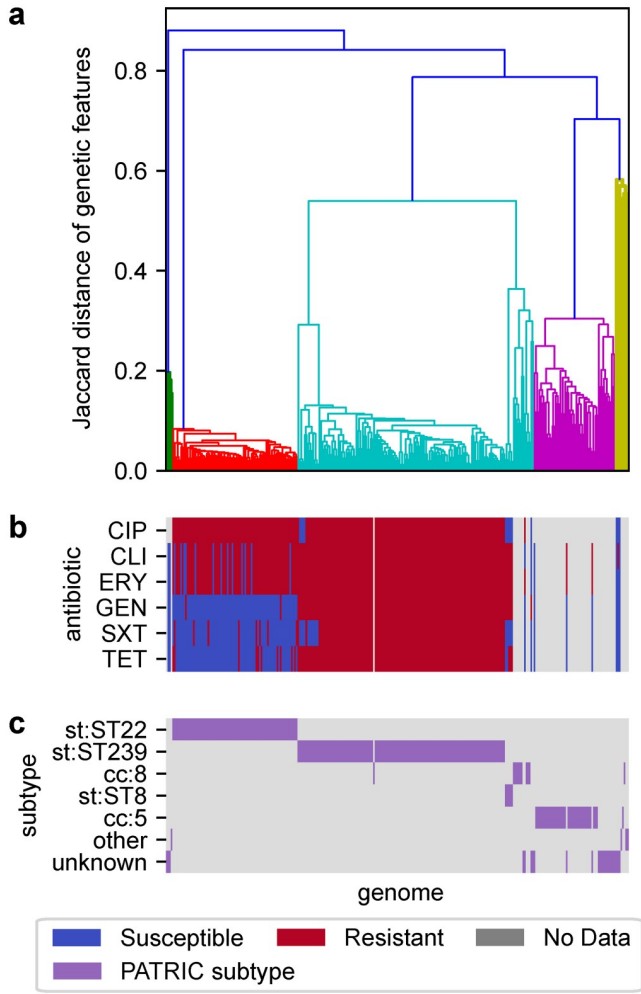

**Fig 1. *S. aureus* genomes clustered by shared genetic content compared to known subtypes and antibiotic resistance patterns.** (a) Genomes clustered using hierarchical clustering with average linkage, based on pairwise Jaccard distances between the sets of genetic features present in each genome. Clusters extracted from this hierarchy align well with (b) experimentally observed resistance patterns and (c) subtype annotations from PATRIC. Antibiotics shown are ciprofloxacin (CIP), clindamycin (CLI), erythromycin (ERY), gentamicin (GEN), sulfamethoxazole/ trimethoprim (SXT), and tetracycline (TET).

hits, though in the cases of clindamycin and erythromycin, no genes were found significant even with a less stringent Benjamini-Hochberg correction to FDR $\leq$ 0.05.

## SVM random subspace ensembles identify known AMR genes in *S. aureus*, *P. aeruginosa*, and *E. coli* across multiple antibiotics

We applied our SVM-RSE approach to identify AMR genes in the larger *P. aeruginosa* and *E. coli* pan-genomes, using the same core allele/non-core gene encoding of genomes and focusing on features positively associated with resistance. In addition to the six *S. aureus* cases, SVM-RSEs were trained to predict resistance for ten more organism-antibiotic cases: for amikacin, ceftazidime, levofloxacin, and meropenem in *P. aeruginosa*, and for amoxicillin/clavulanic acid, ceftazidime, ciprofloxacin, gentamicin, imipenem, and trimethoprim in *E. coli*, for a total of 16 organism-antibiotic cases (S2 Table, Fig 3a).

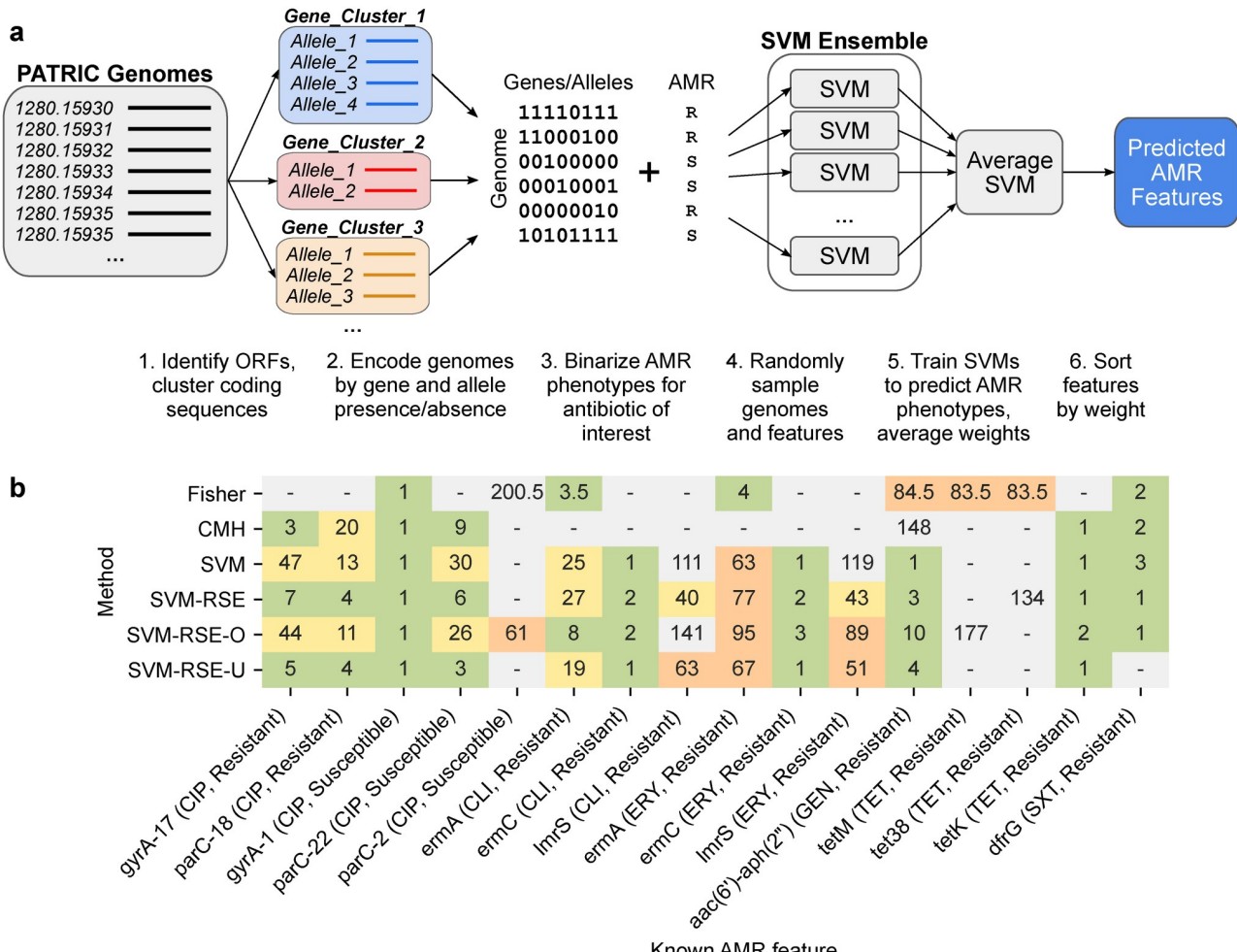

**Fig 2. Comparison of SVM ensemble approaches and statistical tests for detecting AMR-conferring genes and alleles in *S. aureus*.** (a) Workflow for SVM ensemble approaches. Beginning with genomes from PATRIC, open reading frames (ORFs) are identified and clustered by coding sequence to identify putative genes and alleles. Each genome is encoded based on the presence or absence of each gene and allele to capture genomic variation in the pan-genome as a sparse binary matrix. Genomes and/or features of this matrix are randomly sampled 500 times and used to train SVMs to predict binary AMR phenotype for a single antibiotic from genotype. Weights for each feature are averaged across all models in the ensemble and used to rank features by association to AMR. (b) Associations between known AMR-conferring genomic features and AMR phenotype, as ranked by Fisher's Exact test, Cochran-Mantel-Haenszel test, and four different SVM ensemble types (SVM: ensemble by bootstrapping genomes, SVM-RSE: bootstrapping genomes and features; "random subspace ensemble", SVM-RSE-O: SVM-RSE with oversampling to balance subtypes, SVM-RSE-U: SVM-RSE with undersampling to balance subtypes). Features were ranked either by p-value for statistical tests or by average feature weight for SVM ensembles. Fractional ranking was used for ties. Only features detected by at least one method are shown, colored by rank (green: in top 10, yellow: 11–50, orange: 51–100, gray: >100). Features shown are either genes or individual alleles (denoted as <gene>-#).

By examining the highest weighted features in each SVM-RSE, this approach was able to identify known AMR genes among the top 50 hits in 15 out of the 16 cases, with more than half of those hits occurring within the top 10 and at least one known AMR gene found among the top 10 in 13 out of the 16 cases (Table 2). Only in the case of *P. aeruginosa*-amikacin were no such genes found, in which all aminoglycoside-inactivating enzymes in the pan-genome identified by sequence homology had either modest LORs for resistance or were extremely rare (S4 Table). In total, 10, 7, and 28 unique AMR genetic features previously described in literature were detected and associated to the correct antibiotic for *S. aureus*, *P. aeruginosa*, and *E. coli*, respectively.

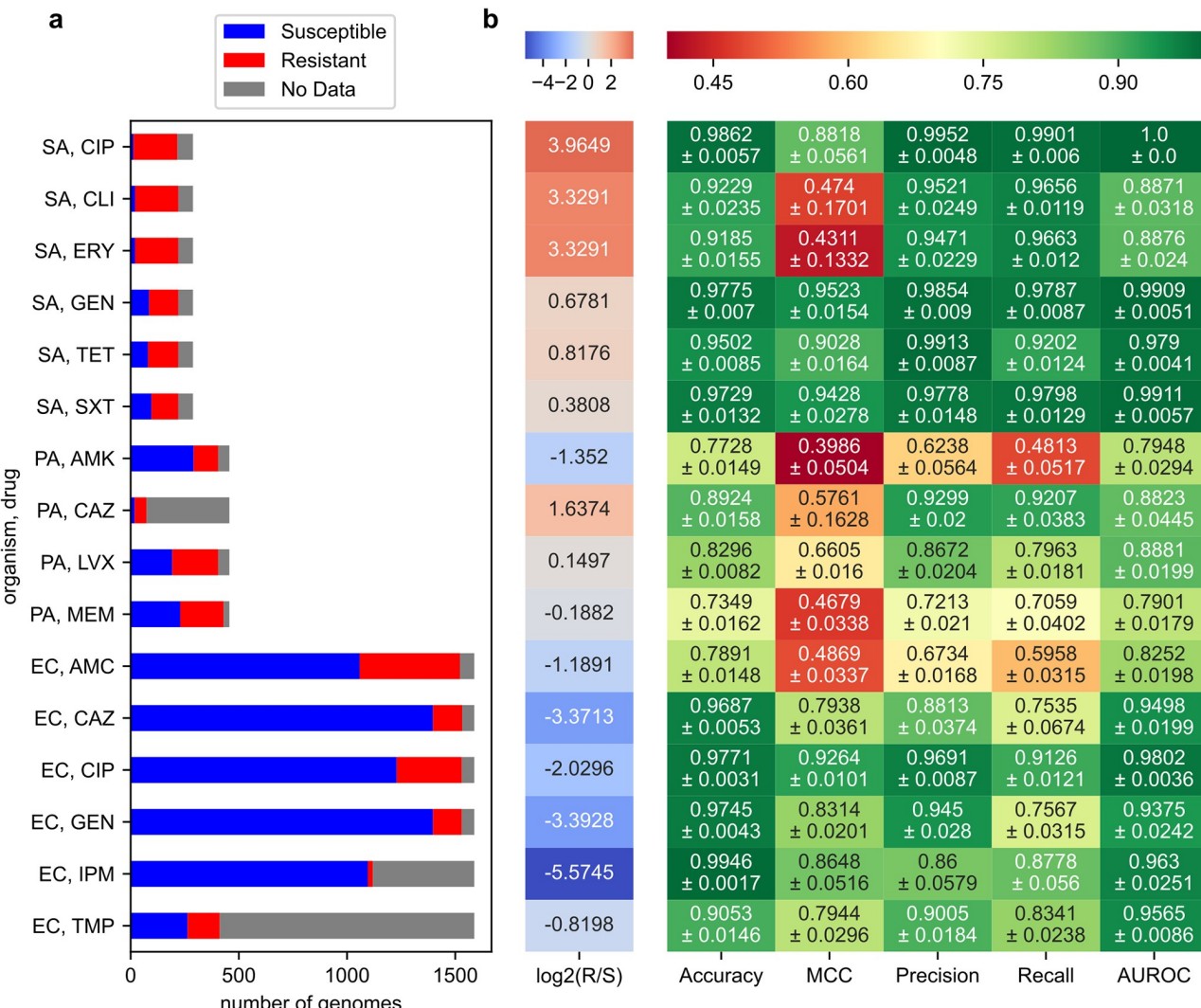

**Fig 3. Predictive performance of SVM-RSE on 16 organism-antibiotic cases.** (a) Distribution of AMR phenotypes for each case. Organisms examined are *S. aureus* (SA), *P. aeruginosa* (PA), and *E. coli* (EC). Antibiotics examined are ciprofloxacin (CIP), clindamycin (CLI), erythromycin (ERY), gentamicin (GEN), tetracycline (TET), sulfamethoxazole/trimethoprim (SXT), amikacin (AMK), ceftazidime (CAZ), levofloxacin (LVX), meropenem (MEM), amoxicillin/clavulanic acid (AMC), imipenem (IPM), and trimethoprim (TMP). (b) SVM-RSE performance metrics from 5-fold cross validation. Performance values shown are averages and standard errors from 5-fold cross validation. The left-most column "log2(R/S)" shows the extent of class imbalance, the log2 of the number of resistant genomes divided by the number of susceptible genomes.

In terms of AMR phenotype prediction, in all 16 cases the individual SVMs of the corresponding SVM-RSE achieved much higher Matthew's correlation coefficients (MCCs) on the test set when trained on the true data compared to data where AMR phenotypes were randomly permuted, suggesting that the associations learned were not due to noise (S3 Fig). As a whole, the SVM-RSE achieved accuracies ranging from 79.3% to 99.5%, MCCs ranging from 0.394 to 0.952, and area under curves (AUCs) ranging 0.790 to 1.0 on the test set when averaged across 5-fold cross validation experiments (Fig 3b, S4 Fig). The average precision and recall ranged from 0.624 to 0.995 and 0.481 to 0.990, respectively (Fig 3b). Across these metrics, 6 of 7 problematic cases were either 1) *P. aeruginosa* cases, which involve a notably larger genome than the other two organisms and thus present more challenging prediction problems, or 2) strongly class-imbalanced cases (*S. aureus*-clindamycin, *S. aureus*-erythromycin), though

**Table 2. Known resistance-conferring genes found by SVM-RSE in *S. aureus*, *P. aeruginosa*, and *E. coli*.**

| Organism | Drug | Features | Ranked 1–10 | Ranked 11–50 |
|---|---|---|---|---|
| *S. aureus* | CIP | 2 | *gyrA* [21,22], *parC* [21,22] | - |
| *S. aureus* | CLI | 3 | *ermC* [24,25] | *ermA* [24,25], *lmrS* [26] |
| *S. aureus* | ERY | 2 | *ermC* [24,25] | *lmrS* [26] |
| *S. aureus* | GEN | 1 | *aac(6′)-aph(2″)* [28,29] | - |
| *S. aureus* | SXT | 1 | *dfrG* [34] | - |
| *S. aureus* | TET | 1 | *tetK* [31] | - |
| *P. aeruginosa* | AMK | 0 | - | - |
| *P. aeruginosa* | CAZ | 1 | - | *muxC* [35] |
| *P. aeruginosa* | LVX | 4 | *gyrA* **(2)** [36], *parC* [36], *oprD* [37] | - |
| *P. aeruginosa* | MEM | 2 | *oprD* [37], *bla*$_{OXA-2}$ [38] | - |
| *E. coli* | AMC | 2 | *bla*$_{OXA-1}$ [39], *bla*$_{TEM}$ [39] | - |
| *E. coli* | CAZ | 4 | *bla*$_{CTX-M}$ [39], *bla*$_{SHV}$ [39], *bla*$_{CMY}$ [39] | *bla*$_{OXA-1}$ [39] |
| *E. coli* | CIP | 8 | *parC* [40], *gyrA* **(4)** [40] | *parC* [40], *parE* [40], *mdtA* [41] |
| *E. coli* | GEN | 6 | *aac(3)-IId/III* [42,43], *ant(2″)-Ia* [43], *ant(3″)-Ia* [42,43] | *aac(3)-VIa* [42,43], *aac(6′)-Ib* [42,43], *ant(3″)-Ia* [42,43] |
| *E. coli* | IPM | 3 | - | *bla*$_{CTX-M}$ [39], *mdtA* [41], *bla*$_{NDM}$ [39] |
| *E. coli* | TMP | 5 | *dfrA1* [44], *dfrA17* [44], *dfrA14* [44] | *qacE* [45], *dfrA12* [44] |

For each organism-antibiotic pair, known AMR genes among the top 50 features detected by SVM-RSE are shown. Features referring to individual alleles of a gene are underlined. In the cases of *P. aeruginosa*-LVX and *E. coli*-CIP, two and four distinct resistant *gyrA* alleles were found in the top 10, respectively. In cases where a gene is mentioned in both the top 10 and rank 11–50 columns, multiple resistant alleles were detected at the different ranks. Antibiotics examined are ciprofloxacin (CIP), clindamycin (CLI), erythromycin (ERY), gentamicin (GEN), sulfamethoxazole/trimethoprim (SXT), tetracycline (TET), amikacin (AMK), ceftazidime (CAZ), levofloxacin (LVX), meropenem (MEM), amoxicillin/clavulanic acid (AMC), imipenem (IPM), and trimethoprim (TMP).

other strongly class-imbalanced cases performed well (*S. aureus*-ciprofloxacin, most *E. coli* cases). The final problematic case of *E. coli*-AMC is reasonably well balanced and may point to the challenge of predicting resistance for combination therapies of drugs with interacting mechanisms. Nonetheless, the models with the highest predictive performance were not necessarily those with the best detection of known AMR determinants and vice versa, which highlights the need for AMR prediction models to be evaluated both in terms of prediction performance and biological relevance.

Finally, we examined whether these top hits are robust to the core gene threshold used to determine which features of the pan-genome are encoded at the gene level and which are encoded at the allele level. Compared to our original threshold of designating all genes missing in no more than 10 genomes as core genes, we also encoded each pan-genome using two relative core gene thresholds: genes missing in no more than 2% or 10% of all genomes. After repeating the SVM-RSE workflow with these alternate pan-genome representations, the set of the top 50 resistance-associated and top 50 susceptibility-associated was reasonably conserved between all thresholds. Across all organism-antibiotic cases, the average Jaccard similarity of selected features was 0.744 when comparing thresholds of 10 vs 10%, and 0.818 when comparing thresholds of 10 vs 2% (S5 Fig).

## Assessment of bias in features selected by SVM random subspace ensembles

We explored two potential biases in the features selected by SVM-RSE: whether there is a preference for genes with low versus high sequence variability, or for chromosomally versus plasmid encoded genes. First, as our approach encodes core genes at the allele level, we examined whether sequence variability impacts the selection of core gene alleles. Within each pan-genome, the number of unique alleles ("allele count") for each core gene was computed, and

for each organism-antibiotic case, the allele count distribution of the genes corresponding to selected core gene alleles was compared to that of all core genes (S6a and S6b Fig). Across all cases, there is a consistent but modest bias towards selecting core genes with higher sequence variability. However, even in the cases with the largest difference in mean allele count, the allele count distribution for selected core features is nearly indistinguishable from that of all core genes (S6c–S6e Fig).

Second, we examined whether SVM-RSE is capable of selecting non-core genes that are plasmid encoded. Contigs from all genome assemblies were identified as plasmid or chromosomal based on similarity to known plasmids on PLSDB [46], and genes with a majority of their alleles located on plasmid contigs were labeled as plasmid encoded genes. For each organism-antibiotic case, the number of selected non-core plasmid and chromosomal genes was compared to that of all non-core genes (S5 Table). SVM-RSE selected plasmid genes in 10/16 cases, with eight cases showing enrichment for plasmid genes. The six cases in which plasmid genes were not selected fall into two categories: 1) involving fluoroquinolones (ciprofloxacin, levofloxacin), for which resistance is primarily mediated by mutations in chromosomal genes *gyrA* and *parC*, or 2) involving *P. aeruginosa*, for which a relatively small fraction of non-core genes could be identified as plasmid encoded (1.3%, compared to 4.1% of *S. aureus* and 3.0% for *E. coli*). Overall, the SVM-RSE approach for identifying AMR-associated genetic features appears to be robust to sequence variability when selecting core gene alleles, as well as sensitive to plasmid genes when selecting non-core genes.

## SVM random subspace ensembles specify the space of *gyrA* and *parC* mutations associated with fluoroquinolone resistance

We examined resistance to fluoroquinolones (FQs) to compare AMR patterns in different organisms against the same drug class. For all three organisms, the SVM-RSE approach successfully detected at least one allele from both of the two established targets of FQs, *gyrA* and *parC*, within both the top 10 resistance-associated genetic features and the top 10 susceptibility-associated genetic features. All *gyrA* and *parC* alleles that the SVM-RSE associated with resistance bore substitutions previously known to confer resistance to FQs, while those that the model associated with susceptibility had no such known mutations (Table 3). Additionally, there were no uncharacterized mutations among the resistance-associated alleles that were not also present in a susceptibility-associated allele, which suggests that FQ resistance attributable to *gyrA* and *parC* may be limited to a narrow space of mutations, even across multiple organisms. Upon examining all *gyrA* and *parC* alleles, we find that resistance conferred by individual *gyrA* alleles is not dependent on a specific *parC* allele or vice versa; the LOR for resistance of any given *gyrA/parC* allele pair is not larger than that of the corresponding *gyrA* or *parC* alleles individually (S7 Fig). By this metric, there were also no strong pairwise epistatic effects apparent between any of the top 10 resistance-associated hits in all three organisms (S8 Fig).

## Characterization of candidate novel AMR genes

In order to reduce the set of top resistance-associated genetic features to a smaller number of higher confidence AMR gene candidates, we filtered the top 10 hits for each organism-antibiotic case based on existing annotations and the level of sequence variability in each hit's assigned gene cluster (see Methods). This yielded 25 candidate AMR-associated features which were further characterized by domain annotations from InterPro [51] (Table 4). In 9 out of the 13 core gene allele candidates, only a subset of the mutations present in the predicted AMR-conferring allele were actually enriched for resistance; those mutations were found to be

**Table 3. Alleles of *gyrA* and *parC* associated with fluoroquinolone resistance detected by SVM-RSE.**

| Organism | Feature | # Res. | # Sus. | Mutations |
|---|---|---|---|---|
| *Alleles associated with fluoroquinolone resistance* | | | | |
| *S. aureus* | *gyrA-18* | 119 | 0 | **S84L** [47,48], D402E, T457A, V598I, Δ815, T818E, Δ824, Δ825, E859V, E886D |
| *S. aureus* | *parC-17* | 113 | 0 | **S80F** [48], F410Y |
| *P. aeruginosa* | *gyrA-4* | 82 | 2 | **T83I** [49,50] |
| *P. aeruginosa* | *gyrA-15* | 18 | 1 | **T83I** [49,50], Δ909, Δ910 |
| *P. aeruginosa* | *parC-2* | 78 | 1 | **S87L** [49,50] |
| *E. coli* | *gyrA-5* | 66 | 1 | **S83L**, **D87N** [22,40] |
| *E. coli* | *gyrA-6* | 15 | 0 | **S83L**, **D87N**, [22,40] D678E, A828S |
| *E. coli* | *gyrA-9* | 157 | 2 | **S83L**, **D87N**, [22,40] A828S |
| *E. coli* | *gyrA-14* | 27 | 0 | **S83L**, **D87N**, [22,40] D678E |
| *E. coli* | *parC-6* | 46 | 2 | **S80I** [22,40] |
| *Alleles associated with fluoroquinolone susceptibility* | | | | |
| *S. aureus* | *gyrA-22* | 2 | 4 | D402E, T457A, V598I, Δ815, T818E, Δ824, Δ825, E859V, E886D |
| *S. aureus* | *parC-1* | 0 | 12 | F410Y |
| *P. aeruginosa* | *gyrA-1* | 23 | 115 | - |
| *P. aeruginosa* | *gyrA-6* | 4 | 39 | Δ909, Δ910 |
| *P. aeruginosa* | *parC-1* | 52 | 137 | - |
| *E. coli* | *gyrA-0* | 3 | 637 | D678E, A828S |
| *E. coli* | *gyrA-1* | 1 | 152 | D678E |
| *E. coli* | *gyrA-22* | 2 | 179 | - |
| *E. coli* | *parC-1* | 1 | 250 | - |
| *E. coli* | *parC-2* | 7 | 475 | D475E |

Alleles of *gyrA* and *parC* among the top 10 hits associated with either resistance or susceptibility by SVM-RSE were characterized based on mutations relative to the corresponding gene in a reference genome for each organism: NC_002745.2 for *S. aureus* (N315), NC_022516.2 for *P. aeruginosa* (PAO1), U00096.3 for *E. coli* (K12 MG1655). Allele-specific mutations are shown, with known resistance-conferring mutations shown in bold and underlined. Each allele's frequency among resistant (Res.) and susceptible (Sus.) genomes are shown.

present in known domains of their corresponding core gene and are strong candidates to be AMR-conferring (Fig 4).

We note that a few of the predicted core gene alleles are of genes previously associated to resistance against the corresponding antibiotic, if not necessarily in the target organism or mechanistically established. For instance, an HflX-like protein is known to confer resistance in erythromycin in *Listeria monocytogenes* through ribosome recycling [52], and it is possible that the *hflX* gene discovered here may similarly confer resistance in *S. aureus*. In *Helicobacter pylori*, *oppD* was found to be significantly induced by gentamicin exposure [53]. For *ahpF*, overexpression is known to increase the minimum inhibitory concentration (MIC) for strepto-mycin (another aminoglycoside) [54], and has also been linked to increased multi-drug resis-tance through increased defense against oxidative stress in *E. coli* [55]. Finally, WP_000664727, probable *repL*, has been associated with the replication of staphylococcal

**Table 4. Novel resistance-conferring gene candidates predicted by SVM-RSE.**

*Predicted AMR-conferring core gene alleles*

| Organism | Drug | Gene | R/S | LOR | AMR mutations | Mutation location(s) |
|---|---|---|---|---|---|---|
| *S. aureus* | ERY | *hflX* | 135/0 | 8.8 | Wildtype | - |
| *S. aureus* | GEN | SA_RS03845 | 134/0 | 13.5 | S409N | ABC transporter-like domain |
| *S. aureus* | GEN | *metS* | 134/0 | 13.5 | T506N, E541K | Anticodon-binding domain |
| *S. aureus* | GEN | *oppD* | 134/0 | 13.5 | S68N, N132K | ABC transporter-like domain |
| *S. aureus* | GEN | *comGD* | 134/0 | 13.5 | D126Y | ComG operon protein 4 family (non-cytoplasmic) |
| *S. aureus* | GEN | *ahpF* | 134/0 | 13.5 | E38D, S44T, N112K, S422N, K448N | Thioredoxin-like DSF; FAD/NAD(P)-binding domain |
| *S. aureus* | TET | *secE* | 131/2 | 8.7 | G60R | C-terminus |
| *S. aureus* | TET | SA_RS11525 | 131/2 | 8.7 | H127Y | - |
| *S. aureus* | TET | SA_RS10745 | 130/2 | 8.5 | K641Q | RNA-binding domain S1 |
| *S. aureus* | TET | *kdpB2* | 130/2 | 8.5 | P26L | P-type ATPase TM DSF |
| *P. aeruginosa* | CAZ | PA5359 | 48/1 | 6.3 | DEL 1–24 | N-terminal signal peptide |
| *P. aeruginosa* | CAZ | PA1414 | 48/1 | 6.3 | DEL 1–33 | N-terminus |
| *P. aeruginosa* | CAZ | PA1942 | 48/1 | 6.3 | DEL 1–32 | N-terminus |

*Predicted AMR-conferring accessory genes*

| Organism | Drug | Accession | R/S | LOR | Predicted protein/features |
|---|---|---|---|---|---|
| *S. aureus* | CLI | WP_000664727 | 71/5 | 0.7 | Plasmid replication protein, RepL |
| *S. aureus* | GEN | WP_000134308 | 134/1 | 11.6 | Acyl-CoA N-acyltransferase, GNAT domain |
| *S. aureus* | TET | WP_031824444 | 123/2 | 7.8 | Replication initiation factor |
| *E. coli* | AMC | WP_097223430 | 26/5 | 3.5 | Bacterial toxin RNase RnlA/LsoA |
| *E. coli* | AMC | WP_000710826 | 26/5 | 3.5 | Antitoxin RnlB/LsoB |
| *E. coli* | AMC | WP_000774834 | 25/11 | 2.4 | Plasmid stability protein StbB |
| *E. coli* | CAZ | WP_001620093 | 13/33 | 2.1 | NagB/RpiA transferase-like, DeoR-type HTH domain, DeoR C-terminal sensor domain |
| *E. coli* | CAZ | WP_000243817 | 82/15 | 7.1 | RmlC-like cupin fold metalloprotein, WbuC family |
| *E. coli* | CIP | WP_001304218 | 262/386 | 3.9 | Nucleoside triphosphate hydrolase, AAA domain |
| *E. coli* | GEN | WP_001330846 | 44/1 | 8.5 | TM protein |
| *E. coli* | IMP | WP_001310177 | 2/25 | 2.0 | PyrBI operon leader peptide |
| *E. coli* | TMP | WP_000082530 | 59/9 | 4.1 | Mercury transport protein MerC |

Selected AMR-conferring core gene alleles and accessory genes predicted by SVM-RSE, for *S. aureus*, *P. aeruginosa*, and *E. coli*. For core gene alleles, genes names and mutations are defined relative to the reference genomes N315 (NC_002745.2) for *S. aureus*, PAO1 (NC_002516.2) for *P. aeruginosa*, and K12 MG1655 (U00096.3) for *E. coli*. The number of resistant (R) vs. susceptible (S) genomes are shown for each feature. Log2 odds ratios (LORs) were computed using weighted pseudocounts to account for zeroes in the contingency table (see Methods for details). Protein features and domains were annotated with either InterPro (for core gene alleles) or InterProScan (for accessory genes). Abbreviations not originally in InterPro annotations are DSF (domain superfamily) and TM (transmembrane).

resistance plasmids [56]. Sequences and annotations for these features, as well as for all top 50 hits for all organisms-antibiotic cases are available in S3 and S4 Datasets, respectively.

## Discussion

As the number of publicly-available genome sequences for bacterial pathogens continues to grow, there is an increasing need to develop computational methods capable of discerning insights about antimicrobial resistance at scale. To leverage these highly diverse, genomic data-sets, we have developed a reference strain-agnostic workflow based on pan-genomes for build-ing robust machine learning models capable of predicting AMR phenotypes as well as identifying their genetic determinants. Our SVM-RSE approach was able to detect known resistance genes in three microbial pathogens (*S. aureus*, *P. aeruginosa* and *E. coli)* more

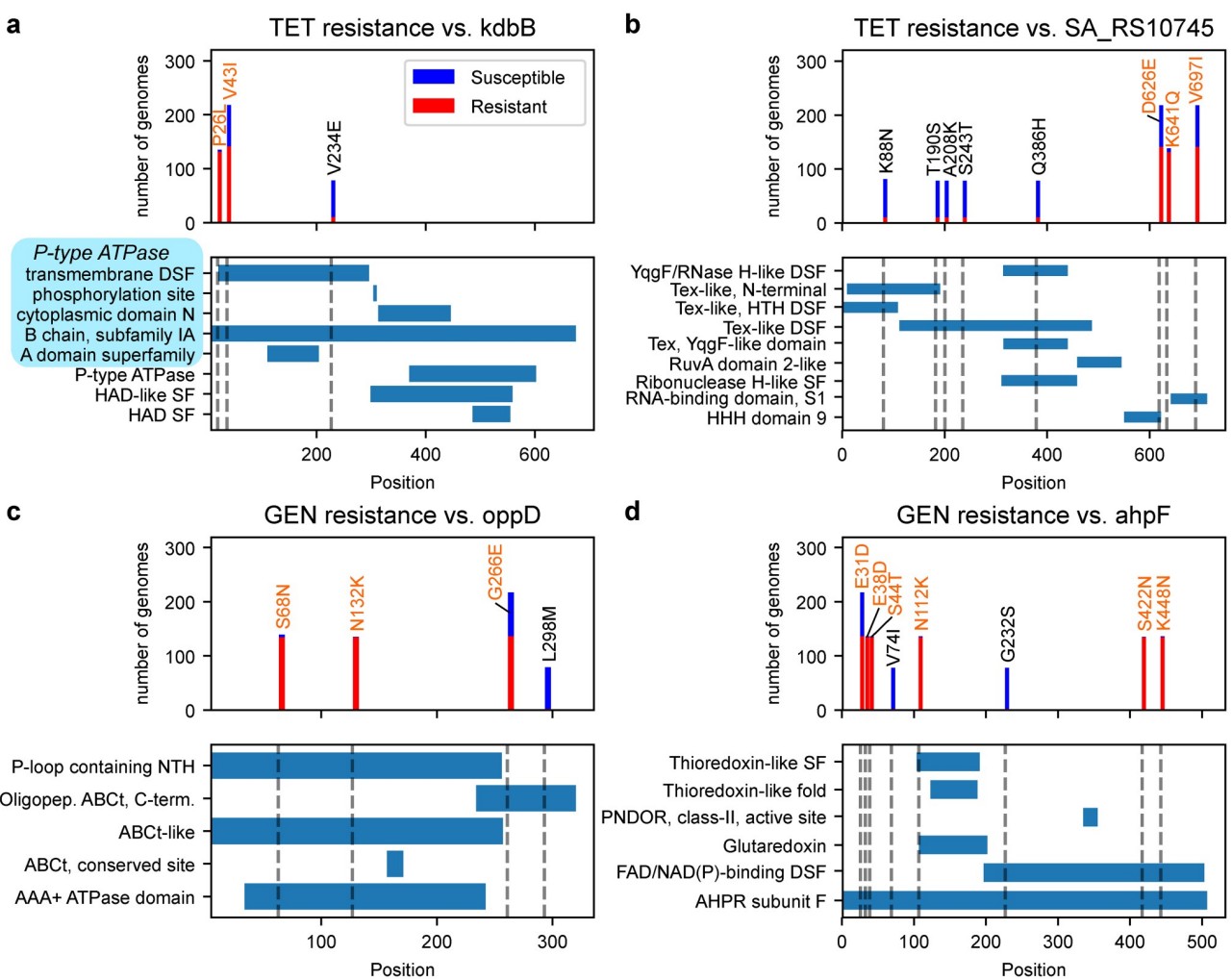

**Fig 4. Characterization of mutations in four predicted AMR-conferring alleles in *S. aureus*.** For each of the predicted AMR-associated genes (a) *kdbB*, (b) SA_RS10745, (c) *oppD* and (d) *ahpF*, the AMR phenotype distributions and locations relative to InterPro structural domains are shown for individual mutations. Mutations in the predicted AMR-associated allele are in orange, while all other mutations observed for that gene are in black (only mutations in at least 5 genomes are shown). For *kdbB*, the first five annotations in light blue are associated with P-type ATPase. Abbreviations include superfamily (SF), domain superfamily (DSF), nucleoside triphosphate hydrolase (NTH), ATP-binding cassette transporter (ABCt), pyridine nucleotide-diphosphate oxidoreductase (PNDOR), and alkyl hydroperoxide reductase (AHPR), in addition to those used in InterPro annotations.

reliably than common association tests, while achieving prediction accuracies competitive with previous machine learning approaches.

Three pan-genomes were constructed from 288 *S. aureus*, 456 *P. aeruginosa*, and 1,588 *E. coli* genomes, and the genetic diversity observed in each species is consistent with what was previously known of each pathogen. Upon integration of AMR profiling data, we found that our SVM-RSE approach effectively identifies established resistance determinants. SVM-RSE detected twice as many known AMR genes than both Fisher's Exact and CMH tests for *S. aureus*, and was able to detect at least one known AMR gene in 15 of 16 organism-antibiotic cases, spanning a total of 45 known AMR associations identified across all three pathogens. Though none of the methods were comprehensive in their detection of all known AMR genes in the pan-genome, the SVM-RSE appears to be the most reliable at detecting those genes for a

diverse array of antibiotic classes. We suspect that the success of this approach may be attributable to the following properties: 1) SVMs by design are capable of capturing structure among multiple features, opposed to independent, bivariate association tests, 2) using an ensemble trained on random genome subsets can more robustly determine important features when the feature set is much larger than the sample set, 3) subsampling features introduces training cases where resistance must be learned without the dominant AMR determinant, which often washes out signal from weaker determinants [3,6], and 4) genes selected by SVM-RSE are neither biased by their extent of sequence variability nor by whether they are plasmid or chromosomally encoded.

The differences in detection rates between cases are partially due to the properties of their corresponding datasets. Generally, more known AMR genes were detected when both a large number of resistant and susceptible genomes were available; the difficult case of *P. aeruginosa*-ceftazidime had only 74 AMR profiles, and cases from the larger *E. coli* dataset typically performed better, with the exception of *E. coli*-imipenem in which only 23 genomes were resistant. In the third problematic case, *P. aeruginosa*-amikacin, AMR profiles were well balanced, but known AMR-conferring genes were rare and/or had modest LORs for resistance, resulting in a more challenging feature selection problem. We also note that while benchmarking with *S. aureus* genomes, the model performed equally well even with aggressive undersampling to evenly represent different lineages. This suggests that genetically "redundant" genomes in a pan-genome may be uninformative with respect to AMR. Finally, in all cases the prediction performances of both individual SVMs and SVM ensembles were high and comparable to previous machine learning approaches, independent of their ability to detect known AMR genes. This result comes as a warning that the raw performance of an AMR-prediction model may have little to do with its capacity to learn real AMR mechanisms.

In a deeper analysis of FQ resistance, we found that the top *gyrA* and *parC* alleles associated with resistance or susceptibility by SVM-RSE segregate perfectly by the presence or absence of known AMR-conferring mutations. The top resistance alleles also bore no uncharacterized mutations that were not also present in susceptible alleles, and no notable epistatic interactions between *gyrA* and *parC* allele pairs or any other pairs of predicted AMR-conferring features could be found. It is possible that the mutational landscape for FQ resistance may be relatively smooth and simple, and FQ resistance may be reliably predicted with simpler techniques; however, such hypotheses will be challenging to validate without more detailed measures of resistance beyond binary AMR phenotypes, such as minimum inhibitory concentrations. Extending this analysis to other predicted hits for all antibiotics identified 25 candidate AMR-conferring genetic features, of which several have evidence in other organisms to be involved in antibiotic-related responses, if not directly contributing to resistance.

Ultimately, by shifting the focus of evaluation from prediction accuracy to biological relevance, our framework more honestly expresses the level of confidence one may have in the generalizability of a machine-learning approach. We find that at the current scale of pathogen sequencing and profiling, our workflow is well-suited for not just predicting AMR profiles, but also identifying genetic features known to confer resistance. The inherent flexibility of this approach opens it up to many improvements to expand the range of biological phenomena the models may draw upon to explain AMR; the incorporation of non-coding genetic features, integration of annotations into the learning process, or implementation of more sophisticated resampling and model aggregation strategies are just a few potential extensions of this work. The continued development of the techniques developed here may eventually be used to systematically extract confident explanations of resistance from pan-genomic datasets to robustly inform responses to the growing AMR threat.

## Materials and methods

### Genome selection and pan-genome assembly

For constructing the *S. aureus*, *P. aeruginosa*, and *E. coli* pan-genomes, genomes on PATRIC [19] were filtered to those that met the following criteria: 1) at least one experimentally measured AMR phenotype (MIC, disk diffusion, agar dilution, Vitek2) is associated with the genome on PATRIC, 2) sequence data is not plasmid-only, and 3) there are at most 100 contigs for *S. aureus* assemblies or at most 250 contigs for *P. aeruginosa* (for *E. coli*, contig filtering was not applied, and only 4 out of 1588 genome assemblies had more than 250 contigs). Genome IDs for selected genomes are available in S1 Dataset. PATRIC genome annotations were used to construct pan-genomes using CD-Hit v4.8.1 [57]. The sequence identity threshold was set at 0.8 and the word length was set to the default of 5.

For each pan-genome, the number of genomes each gene cluster was observed in was computed. The number of core genes was calculated from an increasingly relaxed threshold for core gene, i.e. the maximum number of genomes allowed to be missing a core gene; in all three cases the core-genome size stabilizes by a threshold of 10, which is the threshold used to identify core genes in all subsequent analyses (S1 Fig), and symmetrically to identify unique genes (i.e. genes present in no more than 10 genomes). Within each pan-genome, the unique amino acid sequence variants or "alleles" of each gene were enumerated (S1 Table).

### Mathematical representation of pan-genomes and AMR phenotypes

For each pan-genome, each genome was encoded as a binary vector, based on the presence or absence of every gene cluster and every allele of every gene cluster observed for that organism; this yielded a sparse binary matrix encoding the genetic content at both the gene and allele level (Fig 2a). The number of features was reduced by only analyzing core genes at the allele level, and analyzing non-core genes at the gene level. For each antibiotic, experimental AMR phenotypes were converted to binary vectors by directly converting raw PATRIC AMR annotations "Susceptible" to 0 and both "Resistant" and "Intermediate" to 1. The distribution of binarized phenotypes, typing methods, and typing standards associated with these annotations are in S2 Table.

### Curation of known AMR genes in the *S. aureus* pan-genome

Known AMR genes against antibiotics examined for *S. aureus* were compiled from literature and the CARD database, retrieved on November 26, 2018 [20]. CARD entries were filtered down to those referencing any of the antibiotics examined (ciprofloxacin, clindamycin, erythromycin, gentamicin, trimethoprim, sulfamethoxazole, tetracycline) or their drug classes (fluoroquinolone, lincosamide, macrolide, aminoglycoside, trimethoprim, sulfonamide, tetracycline). Representative protein sequences for these genes were taken from either UniProt or CARD (S2 Dataset) and were aligned to the alleles in the *S. aureus* pan-genome using blastp. Hits with an e-value below $10^{-50}$ and identity >90% were treated as true AMR determinants.

Curated AMR genes were classified into four broad mechanistic categories (S2a Fig): 1) Mutant Site, genes that are direct targets to a given drug that can acquire AMR-conferring mutations, 2) Efflux, genes involved in efflux pumps or regulation of efflux pumps, 3) Modifies Site, genes that protect the direct targets of a given drug, such as by ribosomal modification, and 4) Modifies Drug, genes that cleave, modify, or otherwise inactivate the drug molecule. The frequency and LOR for alleles of curated AMR genes were plotted (S2b and S2c Fig). As most such alleles were very rare and observed AMR phenotypes for many drugs were highly biased towards resistant cases, a modified form of LOR with weighted pseudocounts was

computed to more accurately capture the extent of enrichment and address frequent zeroes in contingency tables

$$LOR = \log_2\left(\frac{\left(AR + \frac{R}{R+S}\right)\left(NS + \frac{S}{R+S}\right)}{\left(AS + \frac{S}{R+S}\right)\left(NR + \frac{R}{R+S}\right)}\right), \quad \begin{array}{l} R = AR + NR \\ S = AS + NS \end{array}$$

where AR is the number of resistant genomes with the allele, AS is the number of susceptible genomes with the allele, NS is the number of susceptible genomes without the allele, and NR is the number of resistant genomes without the allele. This adjustment has the following properties: 1) an allele that is not observed (AR = AS = 0) has a non-informative LOR of 0, 2) a universal allele observed in all genomes (NS = NR = 0) has a non-informative LOR of 0, and 3) the total adjustment to the contingency table is 2, which is common for other pseudocounts strategies for addressing contingency tables with zeroes, such as adding 0.5 to all cells.

## Comparison of statistical tests and SVM ensemble models for predicting AMR determinants in *S. aureus*

For the *S. aureus* pan-genome, Fisher's Exact test and Cochran-Mantel-Haenszel's test were applied between each antibiotic and genetic feature. For CMH, genome subgroups were determined through hierarchical clustering on the genetic feature matrix, implemented in SciPy using pairwise Jaccard distances and average linkage; these clusters were found to be consistent with metadata regarding genome subtype (Fig 1). The two smallest clusters were also treated as a single subgroup for CMH testing. Features were filtered based on significance after either a Bonferroni correction (FWER ≤ 0.05) or Benjamini-Hochberg correction (FDR ≤ 0.05) (S3 Table), then ranked by p-value with fractional ranking for ties.

For each antibiotic, four different types of SVM ensembles of 500 SVMs each were trained to predict AMR phenotype from the *S. aureus* genetic feature matrix, using different resampling strategies (Fig 2a). Within an ensemble, each of the 500 constituent models were trained using one of the following sampling strategies:

1. <u>SVM</u>: Random subsets of 80% of genomes.

2. <u>SVM-RSE</u>: Random subspaces with 80% of genomes and 50% of features.

3. <u>SVM-RSE-U</u>: From each hierarchical clustering subgroup, randomly sample n genomes, where n = 80% of the size of the smallest cluster. Randomly select 50% of features.

4. <u>SVM-RSE-O</u>: From each hierarchical clustering subgroup, randomly sample n genomes, where n = 80% of the size of the largest cluster. Randomly select 50% of features.

SVMs were implemented in scikit-learn, using square hinge loss weighted by class frequency to address class imbalance issues. L1 regularization was included to enforce sparsity for feature selection. For each organism-antibiotic case, genomes without AMR phenotype data were ignored. Features were ranked based on the average feature weight across all SVMs in a given ensemble; in cases where features were subsampled, a feature's average weight was calculated from only SVMs that had access to that feature. For each antibiotic, this yielded a list of top hits associated with resistance (largest positive weights/top ranking features) and a list of top hits associated with susceptibility (largest negative weights/bottom ranking features). Both statistical tests and the four SVM ensemble types were compared based on the number and rank of *a priori* curated AMR determinants detected (Fig 2b).

## Application of SVM-RSE to predict AMR determinants in *S. aureus*, *P. aeruginosa*, and *E. coli*

The SVM-RSE approach described earlier was applied to a total of 16 organism-antibiotic cases across the three organisms to identify genetic features associated with AMR from experimentally observed AMR phenotypes (Fig 3a). For each case, after training an SVM-RSE on the organism's genetic feature matrix and antibiotic's AMR phenotype vector, the top 50 hits associated with resistance were assessed for known AMR determinants and verified through a literature search (Table 2). In examining the *P. aeruginosa*-amikacin case, known aminoglycoside-modifying enzymes were identified in the pan-genome using the same process for curating *S. aureus* AMR genes (S4 Table). LORs were computed using the method as for the curated *S. aureus* AMR genes.

To assess the "null" level of predictive performance, another SVM-RSE was trained for each organism-antibiotic case in which AMR phenotypes were randomly shuffled. For both the original and permuted ensembles, the performance of each of their 500 constituent SVMs was evaluated by computing the Matthew's correlation coefficients (MCCs) on out-of-bag samples, or genomes not used for training (S3 Fig). To assess the overall predictive performance of the ensemble, the SVM-RSE approach was treated as a voting classifier, in which the SVM-RSE prediction is the majority prediction of its 500 constituent SVMs. 5-fold cross validation experiments with the SVM-RSE were conducted for each organism-antibiotic case, and the average and standard error of the accuracy, MCC, precision, recall, and area under receiver operating curve (AUROC) for the testing set across all folds were computed (Fig 3b). ROC curves for each fold were also computed (S4 Fig).

## Assessing stability of SVM-RSE selected features for different core gene thresholds

The core genome of each pan-genome was defined using three thresholds: the set of genes missing in 1) no more than 10 genomes (default), 2) no more than 2% of all genomes, and 3) no more than 10% of all genomes. These core gene thresholds were used to encode each pan-genome in terms of its core gene alleles and non-core genes as described earlier, yielding three distinct matrices per pan-genome (i.e. the genome by gene and allele matrix in Fig 2a). The SVM-RSE analysis was repeated for each pan-genome matrix to predict AMR-associated features for all organism-antibiotic cases. The top 50 resistance-associated features and top 50 susceptibility-associated features yielded by each threshold for each organism-antibiotic case were identified and combined into a single top feature set for each threshold; pairs of these top feature sets across different thresholds were compared by identifying what fraction of features were shared (selected under both thresholds), not shared (available under both thresholds but selected in only one), or differentially encoded (available under only one threshold and impossible to be shared) (S5 Fig).

## Assessing enrichment for highly variable genes among selected features

The total number of unique alleles observed for each core gene ("allele count") was computed for each species' pan-genome. For each organism-antibiotic case, the mean and median allele count of core genes for which at least one allele was selected by SVM-RSE to be associated with resistance ("selected core genes") was computed. This was compared to the mean and median allele count of all core genes for each species (S6a and S6b Fig). For each species, the full allele count distribution of selected core genes was compared to that of all core genes for the

organism-antibiotic case with the largest difference in mean allele count between selected and all core genes (S6c–S6e Fig).

## Assessing enrichment for plasmid genes among selected features

To identify which genetic features were located on plasmids, every contig in every genome assembly was compared to known plasmids in PLSDB (version 2019_10_07) [46] using MASH [58] set to a distance threshold of 0.01, i.e. contigs with distance < 0.01 to a known plasmid were marked as plasmid contigs. All alleles found on plasmid contigs and all genes for which a majority of unique alleles were found on plasmid contigs were treated as plasmid features; all other features were treated as chromosomal. For each organism-antibiotic case, the number of plasmid and chromosomal features in the top 50 features selected by SVM-RSE was computed along with the odds ratio for plasmid features with respect to all features for that organism. As plasmid features are predominantly non-core genes, this calculation was also repeated for just non-core features to more accurately reflect enrichment for plasmid features (S5 Table).

## Analysis of *gyrA* and *parC* mutations with respect to fluoroquinolone resistance

The top 10 hits associated with either resistance (highest feature weights) or susceptibility (lowest feature weights) for the *S. aureus*-ciprofloxacin, *P. aeruginosa*-levofloxacin, and *E.coli*-ciprofloxacin cases were filtered down to just alleles of *gyrA* and *parC*. Mutations for these alleles were called relative to the corresponding protein sequence in the following reference genomes: N315 (NC_002745.2) for *S. aureus*, PAO1 (NC_002516.2) for *P. aeruginosa* and K12 MG1655 (U00096.3) for *E. coli*. Individual mutations for these alleles were compared to those known to confer resistance to FQs (Table 3). Across all *gyrA* and *parC* alleles in each pan-genome, the most abundant alleles were selected (top 8 for *S. aureus* and *P. aeruginosa*, top 12 in *E. coli*) and the LOR for resistance to FQ was computed for each allele individually, as well as for each *gyrA*/*parC* pairing to identify potential interactions (S7 Fig). This pairwise interaction analysis was repeated for all pairs between the top 10 hits associated with resistance by SVM-RSE for the three FQ cases (S8 Fig).

## Extracting candidate novel AMR determinants from SVM-RSE weights

For each of the 16 organism-antibiotic cases, the top 10 hits associated with resistance were filtered down to higher confidence candidates for novel AMR determinants using the following steps: Features already known to be associated with AMR were removed. Features annotated as transposases, phage proteins, or other mobile elements were also removed, as their function may be attributable to their position rather than just their presence or sequence. For core gene alleles, mutations were called relative to the corresponding gene in a reference genome (same as in the FQ case study), and only alleles with at least one mutation highly enriched for resistance were kept (>95% of genomes with the mutation are resistant). These mutations were further characterized by their location in predicted domains or other structural features from InterPro (Table 4, Fig 4); only mutations present in at least 5 genomes are shown. For non-core genes, the most common allele of the gene cluster was identified as the dominant allele, and genes with high sequence variability were filtered out to remove noisy gene calls (i.e. cases where >10% of the instances of that gene have an edit distance >10 from the dominant allele). Of the remaining non-core genes, the dominant alleles were annotated using InterProScan [51] and further filtered down to those with at least one domain annotation. LORs for both core gene alleles and non-core genes were computed using the method as for the curated *S. aureus* AMR genes.

## Supporting information

**S1 Fig. Core-genome size for each organism at different core gene thresholds.** For each pan-genome, the threshold for classifying a gene as a core gene was relaxed from allowing at most 0 to at most 50 genomes to be missing the gene. The threshold of 10 genomes used for subsequent analyses is shown.
(TIF)

**S2 Fig. Type and distribution of known AMR genes in the *S. aureus* pan-genome.** (a) Each known AMR gene detected in the *S. aureus* pan-genome was assigned to one of four broad mechanistic categories. For each allele of each known AMR gene, the number of genomes it is present in and the log2 odds ratio (LOR) for resistance against the appropriate drug was plotted, labeled by (b) drug or (c) mechanism.
(TIF)

**S3 Fig. Out-of-bag performance of individual SVMs in each SVM-RSE compared to null models.** For each of the 16 organism-antibiotic cases across (a) *S. aureus*, (b) *P. aeruginosa*, and (c) *E. coli*, the performance of each of the 500 constituent SVMs used in the corresponding SVM-RSE was assessed as the Matthew's correlation coefficients (MCCs) when predicting AMR phenotypes for out-of-bag genomes (those not used for training), shown in blue. The out-of-bag MCCs of constituent SVMs of SVM-RSEs trained using randomly shuffled AMR phenotype annotations are shown in orange.
(TIF)

**S4 Fig. Receiver operating curves of SVM-RSE models from 5-fold cross validation.** ROC curves for each of the 16 organism-antibiotic cases across (a) *S. aureus*, (b) *P. aeruginosa*, and (c) *E. coli*. The dark blue curves are mean ROC curves from 5-fold cross validation, the lighter curves are individual ROC curves corresponding to each fold, and the grayed areas are within one standard deviation of the mean ROC curve.
(TIF)

**S5 Fig. Consistency of selected features for different core gene thresholds.** The top 100 features (top 50 resistance-associated + top 50 susceptibility-associated) were identified using SVM-RSE for three different core gene thresholds (10: missing from at most 10 genomes, 10%: missing from at most 10% of all genomes, 2%: missing from at most 2% of all genomes). For each pair of thresholds, the fraction of shared vs. non-shared features in the union of their top 100 feature sets were computed. Non-shared features were classified as either "not shared", where both representations contain the feature, or "diff. coded", where the feature is only available under one of the thresholds.
(TIF)

**S6 Fig. Overall sequence variability of selected core gene alleles.** For each organism-antibiotic case, the distribution of the number of alleles of all core genes was compared to that of core genes for which at least one allele was selected by SVM-RSE to be associated with resistance or susceptibility. The (a) mean and (b) median of the selected core gene allele count is shown for each case, compared to the mean and median for all core genes of the corresponding species (dotted lines). For each species, the allele count distributions are shown for the case with the largest difference in mean allele count, (c) *S. aureus* vs. sulfamethoxazole/trimethoprim, (d) *P. aeruginosa* vs. amikacin, and (e) *E. coli* vs. ceftazidime.
(TIF)

**S7 Fig. Interactions between *gyrA* and *parC* alleles in fluoroquinolone resistance.** Log2 odds ratios (LORs) for fluoroquinolone resistance were calculated for each *gyrA/parC* allele pairing and compared to individual alleles in (a) *S. aureus*, (b) *P. aeruginosa*, and (c) *E. coli*. Each cell shows the number of resistant genomes with the allele above, the total number of genomes with the allele below, and is colored by LOR; row and column totals do not add up as only the top 8 (for *S. aureus* and *P. aeruginosa*) or top 12 (for *E. coli*) most frequently observed *gyrA* and *parC* alleles are shown. Alleles among the top 10 features detected by SVM-RSE to be associated with fluoroquinolone resistance are in red, while those the SVM-RSE associated with susceptibility are in blue.
(TIF)

**S8 Fig. Interactions between the top model-predicted hits for fluoroquinolone resistance.** For each of the top 10 genetic features predicted by SVM-RSE to be associated with fluoroquinolone resistance in (a) *S. aureus*, (b) *P. aeruginosa*, and (c) *E. coli*, log2 odds ratios (LORs) for resistance were computed for each feature individually as well as for every top feature pairing. Each cell shows the number of resistant genomes with the allele above, the total number of genomes with the allele below, and is colored by LOR. Gene features are denoted by either their gene name, reference genome locus tag, or "Cluster_#" in cases the coding sequence could not be confidently mapped to a known gene. Allele features are denoted as "gene name-allele number". Features known to confer resistance are in red.
(TIF)

**S9 Fig. Comparison of gene frequency, diversity, and functional distributions in the *S. aureus*, *P. aeruginosa*, and *E.coli* pan-genomes.** (a) Distribution of genes categorized by frequency within each pan-genome: i) core: present in all genomes, ii) near-core: missing from at most 10 genomes, iii) accessory: missing from >10 genomes and present in >10 genomes, iv) near-unique: present in 2–10 genomes, v) unique: present in exactly 1 genome. (b) Estimation of pan-genome openness using Heap's Law. The total number of genes (pan-genome size) and number of genes in all genomes (core genome size) was computed as genomes were introduced sequentially from either the *S. aureus* (SA), *P. aeruginosa* (PA), or *E. coli* (EC) pan-genome. Each value represents the median from 2000 random permutations of genome order. The new gene rate (NGR) was fitted to Heap's Law, in which a more negative exponent represents a more closed pan-genome. (c) Log2 odds ratios (LORs) between individual functional categories and the core, accessory (acc), and unique genomes for each organism individually and combined.
(TIF)

**S10 Fig. Distribution of gene functions in the pan-genomes of *S. aureus*, *P. aeruginosa*, and *E. coli*.** The distribution of gene functional categories based on Clusters of Orthologous Groups (COGs) in the core, accessory, and unique genomes are shown, either (a) including, or (b) excluding the "S: Function unknown" category.
(TIF)

**S11 Fig. Distribution of gene functions for different thresholds for core and unique genes.** For each organism, the set of genes in the (a) core genome was assembled for different core gene thresholds (the maximum number of genomes allowed to be missing a core gene), and (b) analogously for unique genes comprising the unique genome (the maximum number of genomes allowed to carry a unique gene). The "S: Function unknown" functional category is not shown.
(TIF)

**S1 Table. Number of core, accessory, and unique genes and alleles in the pan-genome of each organism.**
(DOCX)

**S2 Table. AMR phenotypes of PATRIC genomes and corresponding typing methods and standards.**
(DOCX)

**S3 Table. Number of significant features associated with antimicrobial resistance in *S. aureus*, as detected by Fisher's exact tests and Cochran–Mantel–Haenszel tests.**
(DOCX)

**S4 Table. Aminoglycoside-modifying enzymes identified by sequence homology in the *P. aeruginosa* pan-genome compared to amikacin resistance phenotypes.**
(DOCX)

**S5 Table. Enrichment for plasmid over chromosomally encoded genetic features selected by SVM-RSE.**
(DOCX)

**S6 Table. Comparison of estimates for *S. aureus*, *P. aeruginosa*, and *E. coli* core-genome sizes.**
(DOCX)

**S7 Table. Fisher's exact test p-values between each COG functional category and the combined core, accessory, or unique genomes of *S. aureus*, *P. aeruginosa*, and *E. coli*.**
(DOCX)

**S8 Table. Fisher's exact test p-values between each COG functional category and the individual core, accessory, and unique genomes of *S. aureus* (SA), *P. aeruginosa* (PA), and *E. coli* (EC).**
(DOCX)

**S1 Dataset. PATRIC Genome IDs for *S. aureus*, *P. aeruginosa*, and *E. coli* genomes used in this study.**
(XLSX)

**S2 Dataset. Protein sequences for known AMR-conferring genes relevant to *S. aureus* analysis.** Contains representative protein sequences of genes known to be associated with resistance against ciprofloxacin, clindamycin, erythromycin, gentamicin, sulfamethoxazole, tetracycline, and trimethoprim. Files named <drug>_card_amr.faa contain sequences that were extracted from the CARD database, retrieved November 26, 2018. File other_amr.faa contains additional sequences for AMR-conferring genes from literature and UniProt compiled independent of CARD.
(ZIP)

**S3 Dataset. Protein sequences for the top 50 resistance-associated genetic features identified by SVM-RSE for each organism-antibiotic case.** Files are named <organism>_<antibiotic>_top_hits_seqs.faa, which each contain all protein sequences relevant to the top 50 hits of the corresponding organism-antibiotic case. For selected alleles, the exact protein sequence of the allele is included. For selected genes, the protein sequences of all alleles of that gene observed in the organism's pan-genome are included. The most commonly

observed allele for selected genes is available in S4 Dataset.
(ZIP)

**S4 Dataset. Annotations for the top 50 resistance-associated genetic features identified by SVM-RSE for each organism-antibiotic case.** Includes the following annotation for each genetic feature: 1) ranking from SVM-RSE, 2) the name of the common allele for selected genes, 3) locus tag of the best aligned reference sequence in the corresponding reference genome, if any, 4) gene name of the reference sequence, if available, 5) gene name assigned by eggNOG, if available, and 6) gene functional annotation by eggNOG. Additional details are available in the document.
(XLSX)

**S5 Dataset. Additional figure-associated data.** Contains figure data in tabular format for Figs 1b, 1c, 4, S2b, S2c, S5, S6a, S6b and S9c Figs.
(XLSX)

**S1 Appendix. References for S6 Table.**
(DOCX)

**S1 Text. Supplemental discussion of *S. aureus, P. aeruginosa, and E. coli* pan-genome properties.**
(DOCX)

# Acknowledgments

We thank Dr. Shankar Subramanian for helpful commentary on the evaluation of known resistance genes.

# Author Contributions

**Conceptualization:** Jason C. Hyun, Erol S. Kavvas, Jonathan M. Monk, Bernhard O. Palsson.

**Data curation:** Jason C. Hyun, Jonathan M. Monk.

**Formal analysis:** Jason C. Hyun.

**Funding acquisition:** Jonathan M. Monk, Bernhard O. Palsson.

**Investigation:** Jason C. Hyun.

**Methodology:** Jason C. Hyun, Erol S. Kavvas, Jonathan M. Monk.

**Project administration:** Jonathan M. Monk, Bernhard O. Palsson.

**Resources:** Jonathan M. Monk, Bernhard O. Palsson.

**Software:** Jason C. Hyun, Jonathan M. Monk.

**Supervision:** Erol S. Kavvas, Jonathan M. Monk, Bernhard O. Palsson.

**Validation:** Jason C. Hyun, Erol S. Kavvas, Jonathan M. Monk.

**Visualization:** Jason C. Hyun.

**Writing – original draft:** Jason C. Hyun.

**Writing – review & editing:** Jason C. Hyun, Erol S. Kavvas, Jonathan M. Monk, Bernhard O. Palsson.

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
