## [Decision Letter · Decision Letter 0]

9 Oct 2019

Dear Dr Monk,

Thank you very much for submitting your manuscript 'Machine learning with random subspace ensembles identifies antimicrobial resistance determinants from pan-genomes of three pathogens' for review by PLOS Computational Biology. Your manuscript has been fully evaluated by the PLOS Computational Biology editorial team and in this case also by independent peer reviewers. The reviewers appreciated the attention to an important problem, but raised some substantial concerns about the manuscript as it currently stands. While your manuscript cannot be accepted in its present form, we are willing to consider a revised version in which the issues raised by the reviewers have been adequately addressed. We cannot, of course, promise publication at that time.

Sincerely,

Nicola Segata

Associate Editor

PLOS Computational Biology

Alice McHardy

Deputy Editor

PLOS Computational Biology

[LINK]

Reviewer's Responses to Questions

**Comments to the Authors:**

Reviewer #1: In this submitted manuscript the authors (Hyun et al.) propose a method that predicts antimicrobial resistant (AMR) genes using SVM ensembles trained using random subspace ensemble (RSE). The SVM-RSE predictors were built for 3 of the main multi-drug resistant Bacterial species and 10 antibiotic classes extracting features from the 3 pan-genomes.

The authors show that the SVM-RSEs are more accurate in predicting AMR genes than well-established association tests. Furthermore the technique is reference-agnostic and has the potential to be improved further.

The manuscript is generally nice to read and has ample supporting material.

Following is a list of minor issues that I encountered reading the manuscript:

I) In lines 229-231 the authors state that oversampling or undersampling did not affect the performance. this does not seem so clear from Fig3b, were ranks for SVM-RSE-O and SVM-RSE-U are sensibly higher than those given from SVM-RSE. Please rephrase accordingly or explain in more details.

II) In figures (main and supplementary) change "# of X" with "Number of X"

III) In Fig. 1 it would be nice to have a more descriptive legend (e.g. near-core: missing from <= 10 strains)

IV) In Fig2a, specify the distance used in the y-axis

V) In Fig2b and c, add a legend for the resistance patterns

VI) Fig. S2, and S3: add "Functional category" above the legends

VII) Lines 173-175: there is probably a missing/wrong word in the sentence (e.g. "or are involved efflux span", please fix it.

VIII) Fig 3b: in the x-axis change "ImsR" with "ImrS"

Reviewer #2: In this work, Hyun et al. have developed a Support Vector Machine (SVM)-based method to identify and predict antimicrobial resistance determinants from (pan)genomic data.

First, they build pangenomes for three species of interest (S. aureus, P. aeruginosa and E. coli) and then they use such gene sets to train and apply models for AMR determinants search.

They succeed in the identification of known determinants and their method seems to outperform common association tests and to achieve prediction accuracies competitive with

previous machine learning approaches (although a proper and clear benchmarking is missing in the current submission). The paper is well written and the amount of data analyses is notable.

Overall I have a few concerns regarding the structure of the paper and the lack of experimental validation on some of the predictions.

- The first part of the results is devoted to “standard” pangenomic analyses. It is no news that core and accessory gene sets do not show interesting and significantly enriched COG categories. Those categories are to broad, especially for the purpose of the current analysis. I would suggest to move the first paragraph (“Analysis of genetic diversity in S. aureus, P. aeruginosa, and E. coli pan-genomes”) to SM and start the Results section from the current “Support vector machine ensembles identify known AMR genes more…”.This would allow going straight to the point.

- It is interesting that the SVM-RSE method identifies novel putative AMR determinants. This, however, calls for an experimental validation of the capability of a few of such genes (ideally one for each of the species) to confer resistance to the organism harboring them. This would render the manuscript and the implemented approach much more reliable. Do those substitutions confer resistance to otherwise susceptible strains? Additionally, hflX or oppD are ideal targets to validate the predictions made by the authors.

- I have a (minor) concern about the criterion used to classify a gene into core, near-core, accessory, unique or near-unique. The authors have shown a threshold based on a fixed number of strains (10, page 18, from line 410). However, the genome datasets are highly variable among the selected species (288, 456, and 1588 publicly-available genomes for S. aureus, P. aeruginosa, and 96 E. coli, respectively) so I think It would be worth replacing such a fixed threshold with a relative amount (10%?). I would be curious to see whether the numbers and the results would change applying such a relative criterion.

Reviewer #3: Hyun and colleagues provide a novel machine learning-based approach for finding enriched sets of alleles involved in AMR. These are compared to known AMR gene alleles, and some discussion is provided for the genes that are implicated by the models that were previously unrecognized in these species. The paper is well written and I enjoyed reading it. However, there were several places in the manuscript where I became very confused. In my opinion text requires a major revision that centers on clarifying certain points, and directly focusing on the message of the paper. That being said, I think that it is scientifically sound and that this paper will be of importance to readers.

Here are my comments:

In the intro, has this RSE approach been used with success elsewhere? Was the choice inspired in some way or invented by the authors?

The last paragraph of the intro, line 88, says that RSEs aggregate classifiers trained on random subsets of both the sample set and the feature set. This requires a sentence explaining what the sample set and feature set are in this context. At first I thought samples were genomes, but since this is a study of pan genomes that was not clear (at this point in the paper).

Figure 1a says strains, but I think the authors mean genomes since two genomes of the same strain could exist in the training set.

Did the authors do anything to account for the phylogenetic distribution of the strains of each species?

Step 4 of figure 3a says “Train SVMs on AMR phenotypes, average weights”. I can’t find exactly how the authors processed the AMR phenotypes. In the case of MICs, are they predicting MICs? Was everything converted to susceptible vs. resistant, S vs. I vs. R, susceptible vs. non-susceptible etc. If so, did they use CLSI or EUCAST?

Similarly, this looks like they built an ensemble of SVMs for predicting the key features for one antibiotic at a time, analyzing one species at a time, but that also is not clear until you get very deep into the paper and it is not made clear in the methods.

For any core gene, each mutational variant observed would create another dimension in the vector. Some important core genes would be expected to have very few variants and hence dimensions, and some might have many. Do the authors have any insights on how this might affect accuracy or the ultimate selection of AMR-related features in either a regular SVM or a RSE? Are the important genes, simply those that have few variants?

Coloring and/or scale should be defined in Figure 1c.

The authors use the term allele, throughout, which I believe means alternative forms of the same gene. However, on line 166 they jump to protein space by defining alleles based on blastp. Is the feature vector based on proteins or genes? I think it is proteins. They should clarify, and I recommend careful use of terms here. This also makes Table 3 a bit confusing because I think the table is describing nucleotide SNPs in column 2 (ie. gene space), but I don’t understand how the model is identifying a nucleotide here if it is based on proteins.

Along these lines, the results section starting on line 181 sets up a comparison of statistical tests vs. the SVM method. Some lead in is required here. I think that the authors are performing statistics over the same matrix that they are performing the svm analysis but that isn’t clear in the text.

Based on my recollection, I think that set of antimicrobials and their associated alleles for the 3 species tends to skew toward chromosomally encoded AMR genes (Lines 162-165). Is that the case? I don’t think that action is required in terms of the experimental approach but that should be mentioned, with some discussion of implications, if it is indeed true.

In figure 2A and B I don’t understand how the data can cluster if there is no information on susceptible vs. resistant phenotypes (2B) for many of the strains in the purple clade on the right-hand side of the plot. Were the purple strains used in the model?

The rationale for the computation of pan-genome openness (line 120) and its depiction in Figure 1 and the supplemental figure are not entire clear to me. If it is required, please clarify. If not, it might help the paper if this went in the supplement. This is also true of the extensive examination of overlap in function between core and accessory genes. This is important but not of primary importance, unless I have missed something.

The authors previously wrote a really nice paper on M. tuberculosis (ref 6). Why was this organism excluded from the analysis? Is there something different about this organism that makes it less amenable to the method?

I understand that shifting from an optimization for model accuracy to a focus on gene identification is important, but the accuracy of the models should still be described in greater detail. It only appears on line 265. I think that the precision vs. recall data are still important to this study because they will provide information that the models are correctly balanced and capable of predicting both classes accurately. This in turn lends support to the idea that we should believe the feature list that has been implicated in AMR. To this end, I think that Figure S4 (counts of SR genomes) and S7 (Accuracy vs MCC) should be brought back into the main text, and that either precision and recall, or F1 should be showed as well.

It seems like this method casts a wide net. That is, the top features are enriched by known AMR variants, but that there is still a lot of genes that are being pulled in and that some analysis is required to determine their biological relevance. It would be nice to get these lists. It would bring interest to the bench scientists.

Small things:

Line 154 define DHFR

Line 249, I think the authors mean trimethoprim/sulfamethoxazole, but I’m not 100% sure.

Line 265 “an accuracies”

In Dataset S1, the underscores in the PATRIC ids should be converted to periods, or a table legend is required. Might be good to convert cells to text format first.

**Have all data underlying the figures and results presented in the manuscript been provided?**

Reviewer #1: Yes

Reviewer #2: No: I couldn't find spreadsheets for most of the figures in this submission (e.g Fig. 1C, 2B ...)

Reviewer #3: Yes

PLOS authors have the option to publish the peer review history of their article (what does this mean?). If published, this will include your full peer review and any attached files.

Reviewer #1: No

Reviewer #2: No

Reviewer #3: No

---

## [Decision Letter · Decision Letter 1]

16 Dec 2019

Dear Dr Monk,

We are pleased to inform you that your manuscript 'Machine learning with random subspace ensembles identifies antimicrobial resistance determinants from pan-genomes of three pathogens' has been provisionally accepted for publication in PLOS Computational Biology.

In the meantime, please log into Editorial Manager at https://www.editorialmanager.com/pcompbiol/, click the "Update My Information" link at the top of the page, and update your user information to ensure an efficient production and billing process.

One of the goals of PLOS is to make science accessible to educators and the public. PLOS staff issue occasional press releases and make early versions of PLOS Computational Biology articles available to science writers and journalists. PLOS staff also collaborate with Communication and Public Information Offices and would be happy to work with the relevant people at your institution or funding agency. If your institution or funding agency is interested in promoting your findings, please ask them to coordinate their releases with PLOS (contact ploscompbiol@plos.org).

Thank you again for supporting Open Access publishing. We look forward to publishing your paper in PLOS Computational Biology.

Sincerely,

Nicola Segata

Associate Editor

PLOS Computational Biology

Alice McHardy

Deputy Editor

PLOS Computational Biology

Reviewer's Responses to Questions

**Comments to the Authors:**

Reviewer #1: The minor issues found in the original version of the manuscript have been solved.

The readability of the manuscript has been considerably improved and we are confident that this scientifically sound manuscript would be of interest of the readers of PLoS Computational Biology.

Reviewer #3: The authors have carefully addressed my concerns. As I said in the initial review, I enjoyed reading it. I'm looking forward to citing this work.

**Have all data underlying the figures and results presented in the manuscript been provided?**

Reviewer #1: Yes

Reviewer #3: Yes

PLOS authors have the option to publish the peer review history of their article (what does this mean?). If published, this will include your full peer review and any attached files.

Reviewer #1: No

Reviewer #3: No

---

## [Editor Report · Acceptance letter]

19 Feb 2020

PCOMPBIOL-D-19-01427R1 

Machine learning with random subspace ensembles identifies antimicrobial resistance determinants from pan-genomes of three pathogens

Dear Dr Monk,

I am pleased to inform you that your manuscript has been formally accepted for publication in PLOS Computational Biology. Your manuscript is now with our production department and you will be notified of the publication date in due course.

With kind regards,

Matt Lyles
